# Using Machine Learning Models to Predict Genitourinary Involvement Among Gastrointestinal Stromal Tumour Patients

## Abstract

Gastrointestinal stromal tumors (GISTs) can lead to involvement of other organs, including the genitourinary (GU) system. Machine learning may be a valuable tool in predicting GU involvement in GIST patients, and thus improving prognosis. This study aims to evaluate the use of machine learning algorithms to predict GU involvement among GIST patients in a specialist research center in Saudi Arabia. We analyzed data from all patients with histopathologically confirmed GIST at our facility from 2003 to 2020. Patient files were reviewed for the presence of renal cell carcinoma, adrenal tumors, or other genitourinary cancers. Three supervised machine learning algorithms were used: Logistic Regression, XGBoost Regressor, and Random Forests. A set of variables, including independent attributes, was entered into the models. A total of 170 patients were included in the study, with 58.8%q (n=100) being male. The median age was 57 (range 9-91) years. The majority of GISTs were gastric (60%, n=102) with a spindle cell histology. The most common stage at diagnosis was T2 (27.6%, n=47) and N0 (20%, n=34). Six patients (3.5%) had GU involvement. The Random Forest model achieved the highest accuracy with 97.1%. Our study suggests that the Random Forest model is an effective tool for predicting GU involvement in GIST patients. Larger multicenter studies, utilizing more powerful algorithms such as deep learning and other artificial intelligence subsets, are necessary to further refine and improve these predictions.

## 1   Introduction

Gastrointestinal stromal tumors (GISTs) are a rare type of mesenchymal tumor that commonly develop in the gastrointestinal tract. In fact, GISTs are the most frequently occurring mesenchymal tumor in this anatomical region [1]. GISTs are known to have several distinct molecular subtypes, including those with mutations in KIT or PDGFRa. Detecting these molecular alterations at an early stage is critical as it can significantly impact the choice of adjuvant and metastatic treatments [2]. Existing literature suggests that GISTs have a nearly equal distribution between genders, with a higher incidence among individuals over the age of 60. Furthermore, GISTs tend to present with symptoms, indicating a symptomatic nature of the disease [3]. Studies conducted in Saudi Arabia have shown that GISTs are predominantly located in the stomach and have a higher incidence in males over the age of 40 years [4]. Although GISTs primarily occur in the stomach and intestine, some patients may experience lower urinary tract symptoms that suggest genitourinary involvement. Additionally, extragastrointestinal stromal tumors of the urinary bladder wall have been observed in rare cases [5]. Currently, an accurate diagnosis of GISTs requires extensive imaging studies, pathological examination, and immunohistochemical analysis [6]. Early diagnosis is imperative

to achieve high rates of disease-free survival, yet the extensive testing required for a diagnosis takes substantial time [7]. Therefore, implementing technology that predicts the involvement of other organs among GIST patients could significantly impact the overall prognosis of this condition. Recent research suggests that using artificial intelligence (AI) and deep learning algorithms may provide more accurate confirmation of the malignant potential of GISTs [8]. The implementation of machine learning techniques, including supervised learning algorithms, has shown promising results in improving the accuracy of predictions for various medical conditions. In this study, we aim to utilize machine learning to predict genitourinary involvement in GIST patients, with a particular focus on the Saudi Arabian population. By utilizing a large dataset of patients diagnosed with GIST from our specialist research center between 2003 and 2020, we aim to determine the accuracy and effectiveness of three supervised machine learning algorithms: Logistic Regression, XGBoost Regressor, and Random Forests. The identification of predictive variables and the accuracy of these models will provide valuable insight into the potential for AI and machine learning to improve the diagnosis and management of GIST patients, particularly in the context of genitourinary involvement.

## 2 Material and Methods

This retrospective study included all patients with a histopathological diagnosis of GIST at King Faisal Specialist Hospital and Research Centre between 2003 and 2020. Any involvement of genitourinary cancer was identified. Data were analysed using SPSS v26. From IBM. Continuous data summarized as mean and standard deviation (SD), whereas categorical data summarized as absolute values and percentages.

Four types of Artificial Intelligence algorithms were employed in this study to predict the presence of genitourinary cancer in the presence of GIST. These include Random Forest, XGBoost Classifier, Catboost classifier and Support Vector Machine. After running a base line prediction model, some variables were dropped because they were not significant to the prediction of the model. The machine learning models were fitted using scikit-learn 0.18 modules of python throughout this study. The data set was randomly divided into the 80% of the training set, and the 20% of the test set at 8:2 (136: 34). The target variable was encoded in a binary format with 1 (presence of genitourinary cancer) and 0 (absence of genitourinary cancer). The RF model is a decision tree-based machine learning model. Each node of the decision tree divides the data into two groups by using a cut-off value inside one of the features. By building an ensemble of randomized decision trees, each of which overfits the data and averages the results to obtain a better classification, the RF technique can reduce the effect of the overfitting problem.

This retrospective chart review study involving human participants followed the standards of the 1964 Helsinki Declaration and its later amendments. This study is a secondary analysis of datasets from an already approved study by the Human Investigation Committee (IRB) and Research Ethics Committee of King Faisal Specialist Hospital and Research Center.

## 3 Results

A total of 170 GIST patients were detected. As shown in Table 1, most of the patients (58.8%; n=100) were males. The median age was 57 (9 to 91) years. The majority of the GISTs were gastric (60%; n=102) with a spindle cell histology. The most common stage at diagnosis is T2 (27.6%; n=47) and N0 (20%; n=34). Six patients (3.5%) had GU involvement. Of them, 3 patients had renal cell carcinomas. two were histologically diagnosed to have clear cell RCC and one with only a radiological diagnosis of RCC. Three other patients had adrenal tumours (one adrenal carcinoma, one isolated adrenal GIST, and one pheochromocytoma).

After all modes of hyper-parameter tuning were done to the model, Random Forest (RF) model achieved the highest accuracy with 97.1%. It predicted that based on the input variables and patient characteristics, 97.1% still did not have associated genitourinary cancer and that only 2.9% of those who had GIST had associated genitourinary cancer. On more analysis to ascertain the specificity of the model, figure 1 shows the confusion matrix for the RF models which explains the specificity of the model in terms of how true the predicted values are accurate to the original values. It showed that out of a random 34 number of patients, the model predicts 32 patients to be GU cancer free even in

Table 1: Demography and Tumour Related Characteristics of Patients (n= 170)

| Part | | |
| --- | --- | --- |
| Continuous variables | n | Med(Range) |
| Age at diagnosis (years) | 170 | $57(9-91)$ |
| GIST Size cm | 161 | $6 (0.3-36)$ |

Table 2: Demography and Tumour Related Characteristics of Patients (n= 170)

| Part | |
| --- | --- |
| Categorical variables | n |
| **Gender** | |
| Male | 100 (58.8) |
| Female | 70 (41.2) |
| **GIST primary site** | |
| Gastric | 102 (60) |
| Small intestine | 47 (27.6) |
| Omentum/peritoneum/mesenteric | 12 (7.1) |
| Other | 9 (5.3) |
| **GIST TNM stage** | |
| T1 | 25 (14.7) |
| T2 | 47 (27.6) |
| T3 | 44 (25.9) |
| T4 | 45 (26.5) |
| N0 | 34 (20.0) |
| N1 | 5 (2.9) |
| M0 | 13 (7.6) |
| M1 | 25 (14.7) |
| **Histopathological subtype** Spindle cell | 85 (50.0) |
| Epithelioid type | 16 (9.4) |
| Mixed epithelioid and spindle | 10 (5.9) |
| Other | 2 (1.2) |

the presence of GIST and only 1 patient to have associated genitourinary cancer in the presence of GIST.

Figure 2 shows the feature importance of each variable column used for the RF model which is the one with the best prediction accuracy. It is evident that variables in the index 5, 3 and 6 contributed more in the prediction. These variables were Associated Cancer taking the highest, Gender and Site of GIST respectively. Therefore, even with the presence of GIST associated cancer, there is rare correlation between GIST and genitourinary cancer.

## 4 Discussion

The study's findings demonstrate the potential of AI technology to accurately predict genitourinary involvement among GIST patients, as evidenced by the RF model's 97.1% accuracy. The patient population analyzed was mostly male. Only a small portion of patients had genitourinary involvement, at less than 5%. The diagnoses for these patients included renal cell carcinoma, adrenal carcinoma, adrenal GIST, and pheochromocytoma

Our study's findings are consistent with existing literature regarding patient demographics and disease characteristics, showing that GISTs are predominantly located in the stomach (61%). The reported age of onset varies across studies, with median diagnosis age ranging from 50 to 60 years [9,10]. However, a study conducted in Saudi Arabia reported a lower mean age at diagnosis of 40 years, which is substantially lower than the median age reported in other studies [4]. Thus, our study's

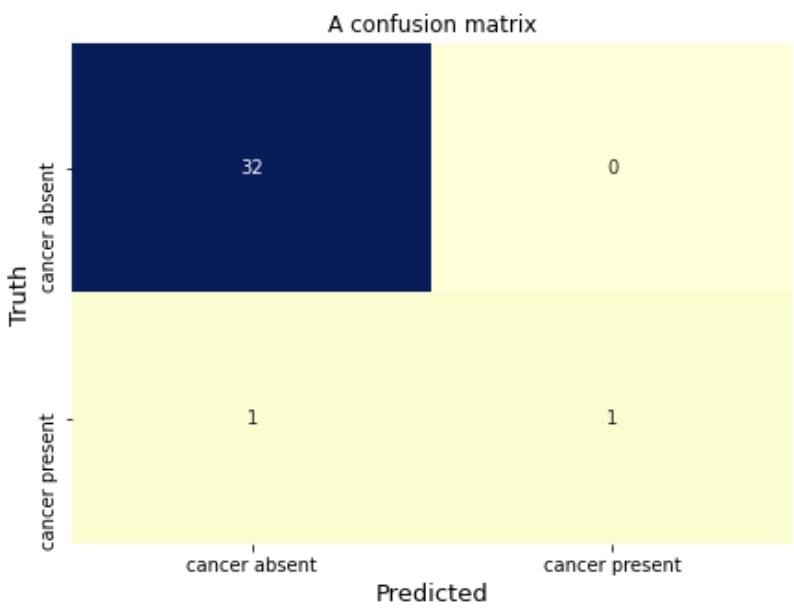

Figure 1: Confusion matrix of the Random Forest model

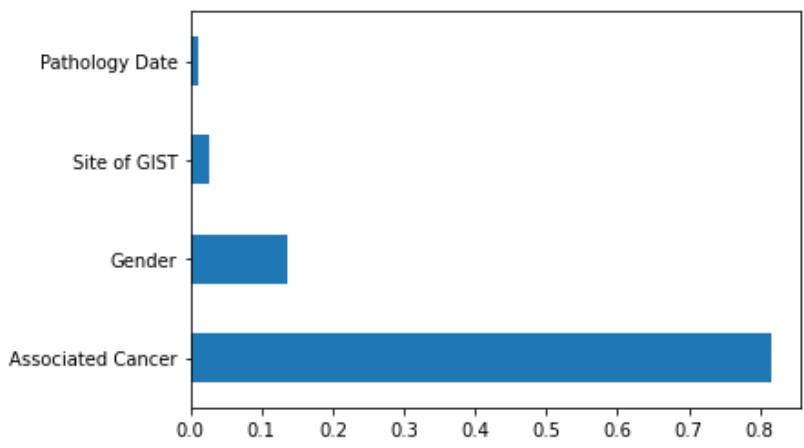

Figure 2: Feature importance of the Random Forest model

results indicate that the age of onset of GIST in our cohort is higher than what has been reported in other studies conducted in Saudi Arabia. This difference may be due to various factors, including differences in sample sizes, selection criteria, and genetic and environmental factors. However, further studies are required to confirm this observation.

This study represents an initial attempt to utilize machine learning algorithms to predict the presence of genitourinary tumors in GIST patients. However, machine learning models have recently been the subject of numerous research studies across various cancer types, including ovarian, thyroid, and breast cancer [11–13]. These studies demonstrate the potential of machine learning in predicting disease outcomes and identifying biomarkers for early diagnosis. Toth et al. demonstrated the successful use of the RF model in clinical practice for the detection of biomarkers for prostate cancer progression. Their study utilized an RF-based classification model to predict aggressive behavior of prostate cancer, achieving an accuracy of 95%. The application of the RF model in their study allowed for the identification of a set of biomarkers that could predict the likelihood of disease progression and guide clinical decision-making [14].

The high accuracy of the RF model in predicting prostate cancer behavior suggests its potential for use in other cancer types, including the prediction of genitourinary involvement in GIST patients as demonstrated in our study. These findings support AI as an externally valid classification model to support the clinical management of prostate cancer [14]. Another study by Xiao et al. reported on similar outcomes predicting the occurrence of prostate cancer using the RF algorithm. Here, transrectal ultrasound findings, age, and serum levels of prostate-specific antigen were taken into account, yielding a predictive accuracy of 83.10%. The results of this study permitted the statement that the adoption of an RF model and AI technology demonstrates superior diagnostic performance than individual diagnostic indicators alone [15]. This is supported by the findings of the present study.

## 5 Limitation and Conclusion

There are several limitations worth noting in this study. Firstly, we did not include all potential predictive factors for genitourinary involvement in GIST patients, such as family history of malignancy and exposure to risk factors. Secondly, this study was conducted at a single center, which may limit the generalizability of our results to other populations. Thirdly, there are currently no other studies in the literature that explore the use of machine learning to predict synchronous GU tumors and GISTs, which makes it difficult to compare and validate our findings. Future research should aim to address these limitations by exploring whether incorporating additional predictive factors into the RF model can improve its accuracy.

This research work can serve as baseline for many future work in exploring the use of state-of-the-art Artificial Intelligence tools, more specifically, machine learning in improving healthcare delivery specifically for cancer patients as early prognosis leads to a better quality of life.

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
