# OpenReview forum: "Using Machine Learning Models to Predict Genitourinary Involvement Among Gastrointestinal Stromal Tumour Patients"
_ICLR.cc/2024/Conference — Submitted to ICLR 2024_

### Official Review · Reviewer_fnHf · 2023-10-21

**Soundness:** 1 poor
**Presentation:** 1 poor
**Contribution:** 1 poor
**Rating:** 1
**Confidence:** 5

**Summary:**

This paper evaluates the use of machine learning algorithms to predict GU involvement among GIST patients in a specialist research center in Saudi Arabia. A total of 170 patients were included in the study, and traditional machine-learning models were applied (Logistic regression, XGBoost, and Random Forests). The Random Forest model achieved the highest accuracy with 97.1%.

**Strengths:**

1) Readability. The paper is simple to read. The problem is clearly explained.

2) Novelty of the application. Few papers applied machine-learning models to this type of problem.

**Weaknesses:**

The paper has several weaknesses and limitations related to the application of machine learning models, from data description to the application of the models. I could not understand the positive and negative class. In terms of model application, I could not find the hyperparameters used in this paper (either a search of them). There is also a lack of results, considering that not all of them are shown in the paper. Also, you have to justify your data separation. Considering that is a small dataset, it is better to use k-fold cross-validation.

In general, the paper must be largely improved by describing all the processes that allow me to observe that machine learning models were applied correctly.

**Questions:**

Not applicable.

**Details Of Ethics Concerns:**

It seems that this paper was submitted to NIPS, too. At the bottom of each page it says: "Submitted to 37th Conference on Neural Information Processing Systems (NeurIPS 2023)." I do not known if the author decision as been submitted.

---

### Official Review · Reviewer_GWdB · 2023-10-28

**Soundness:** 1 poor
**Presentation:** 1 poor
**Contribution:** 1 poor
**Rating:** 1
**Confidence:** 5

**Summary:**

This paper does not adhere to the ICLR format and is not aligned with the area of interest of ICLR. This work should have been desk-rejected.

**Strengths:**

This paper does not adhere to the ICLR format and is not aligned with the area of interest of ICLR.

**Weaknesses:**

This paper does not adhere to the ICLR format and is not aligned with the area of interest of ICLR.

**Questions:**

None.

---

### Official Review · Reviewer_Seo4 · 2023-10-29

**Soundness:** 1 poor
**Presentation:** 1 poor
**Contribution:** 1 poor
**Rating:** 1
**Confidence:** 5

**Summary:**

The paper uses logistic regression, XGBoost and random forest to predict GIST.

**Strengths:**

The paper works on a real-world dataset.

**Weaknesses:**

The paper is at a very immature stage. Here are some detailed comments:
   -- Lack of validation. The test set only contains test samples
   -- No contribution. The paper only presents several supervised learning algorithms on a dataset.

**Questions:**

Why there are only 2 cancer patients in the test set according to Figure 1?

---

### Official Review · Reviewer_UWW5 · 2023-11-01

**Soundness:** 2 fair
**Presentation:** 1 poor
**Contribution:** 1 poor
**Rating:** 1
**Confidence:** 5

**Summary:**

This work uses machine learning models to predict genitourinary system involvement in patients with gastrointestinal stromal tumors. Employing various supervised learning methods including logistic regression, XGBoost, and Random Forest across a dataset of 170 patients from a specific hospital, the researchers achieved impressive accuracy rates around 97%, showcasing the potential of these models in medical classification problems.

**Strengths:**

This paper identifies an important understudied problem in the medical literature. It provides a good motivation/explanation of the underlying issues related to gastrointestinal stromal tumors and the need for ML in this context.

**Weaknesses:**

- *Insufficient Data*: The work uses 170 example and a highly imbalanced distribution of the positive class, consisting of only 6 positive samples. This limitation significantly constrains the ability to train and validate the machine learning models effectively.
- *Class Imbalance and Evaluation Metric*: The severe class imbalance in the dataset makes accuracy a poor choice for an evaluation metric, as it can provide a misleadingly high performance measure.
- *Limited Technical Contribution and Originality*: The application of common machine learning methods without any notable innovation or significant technical advancement limits the paper's contribution to the broader machine learning community.

**Questions:**

N/A

---

### Meta-Review · Area_Chair_36Fp · 2023-12-10

**Metareview:**

There is a clear consensus that this contribution is not significant enough to be appropriate for ICLR.

**Justification For Why Not Higher Score:**

Reviewers are unanimous.

**Justification For Why Not Lower Score:**

Reviewers are unanimous.

---

### Decision · Program_Chairs · 2024-01-16

Reject